# A flexible catheter-based sensor array for upper airway soft tissues pressure monitoring

Jiang Shang[1,5], Xiaoxiao Ma[1,5], Peikai Zou[1], Chenxiao Huang[1], Zhechen Lao[1], Junhan Wang[1], Tingshu Jiang[2], Yanzhe Fu[1], Jiebo Li[1], Shaoxing Zhang[3], Ruya Li[1]✉ & Yubo Fan[1,4]✉

Obstructive sleep apnea is a globally prevalent concern with significant health impacts, especially when coupled with comorbidities. Accurate detection and localization of airway obstructions are crucial for effective diagnosis and treatment, which remains a challenge for traditional sleep monitoring methods. Here, we report a catheter-based flexible pressure sensor array that continuously monitors soft tissue pressure in the upper airway and facilitates at the millimeter level. The sensor's design and versatile 3D femtosecond laser fabrication process enable adaptation to diverse materials and applications. In vitro testing demonstrates high sensitivity (38.1 Ω/mmHg) and excellent stability. The sensor array effectively monitors distributed airway pressure and accurately identifies obstructions in an obstructive sleep apnea animal model. In this work, we highlight the potential of this catheter-based sensor array for long-term, continuous upper airway pressure monitoring and its prospective applications in other medical devices for pressure measurement in human body cavities.

Sleep is a fundamental component of human life, significantly impacting physical, mental, and social well-being. Obstructive sleep apnea (OSA), characterized by recurrent upper airway collapse during sleep, disrupts normal breathing and lowers blood oxygen levels[1–3]. It has been estimated that nearly one billion people have OSA worldwide based on obesity rates, race/ethnicity data, age, and regional country prevalence data, while the vast majority are still undiagnosed[3,4]. This condition presents significant health risks, including cardiovascular diseases[5,6], hypertension[7,8], and cognitive impairments[9,10], emphasizing the pressing need for accurate detection and effective management strategies[2,11]. Precise identification and localization of obstructions are crucial for OSA diagnosis and treatment planning. It is particularly important to assess the obstruction site preoperatively to plan the surgery appropriately for the procedure such as uvulopalatopharyngoplasty (UPPP), in which the obstruction site would significantly affect the operation success rate, 52% with palatal obstruction and 5% with hypopharyngeal obstruction[12].

While polysomnography (PSG) is the gold standard for OSA diagnosis, capturing various sleep-related physiological parameters, it falls short in obstruction localization[1,13]. Various advanced methods such as flexible laryngoscopy[14,15], CT Imaging[16], acoustic pharyngometry[17,18], rhinometry[18,19], cine-MRI[20,21], Drug-Induced Sleep Endoscopy (DISE)[22–24], and pharyngoesophageal manometry[25] are currently being used in the clinical diagnosis of OSA to improve accuracy by measuring upper airway obstruction location[1,2]. However, the testing results from these measurements, conducted in awake or

[1]The Key Laboratory for Biomechanics and Mechanobiology of Ministry of Education, Beijing Advanced Innovation Center for Biomedical Engineering, School of Biological Science and Medical Engineering, Beihang University, Beijing, PR China. [2]Department of Respiratory and Critical Care Medicine, Yantai Yuhuangding Hospital, affiliated with the Medical College of Qingdao University, Yantai, Shandong, PR China. [3]Department of Otolaryngology, Peking University Third Hospital, Beijing, PR China. [4]School of Engineering Medicine, Beihang University, Beijing, PR China. [5]These authors contributed equally: Jiang Shang, Xiaoxiao Ma. ✉e-mail: liruya@buaa.edu.cn; yubofan@buaa.edu.cn

induced sleep states, may differ from natural sleep conditions due to muscular and neural influences. To access obstruction locations during sleep states, there are approaches of employing a catheter-based air pressure sensor array positioned within the human upper airway and detecting gas pressure differences along the airway to locate obstructions regions[26–30]. However, these catheters have limited sensing spatial resolution, so they cannot accurately determine the precise location. Fiber Bragg grating (FBG) has also been used to characterize upper airway obstruction in OSA[31], but the flexibility of optical fiber protective coating and device temperature drift significantly impact the precision of its measurement in the application. Thus, a device that can precisely locate OSA obstructions and conduct precise evaluation and diagnosis would be a significant advancement in the field.

The emergence of flexible electronics has garnered significant interest due to their potential in healthcare monitoring applications[32,33]. In contrast to conventional rigid silicon-based electronics, flexible electronics exhibit notable advantages in enduring diverse mechanical conditions such as tension, compression, bending, and torsion[34,35]. For OSA, the dynamic nature of soft tissue collapse in the upper airway necessitates a sensor capable of detecting pressure from all directions. Flexible pressure sensors integrated catheters, which emerged from flexible electronic technology, offer a potential solution to this challenge[36,37]. Pressure sensors integrated into catheters have been used in several studies to monitor soft tissue pressure in human body cavities, such as blood vessels[38,39], urinary system[37,40–42], and gastrointestinal tract[43–46]. These sensor-integrated catheters have a shape that bends according to the angle of the airway and a flexible wall that records pressure from all directions. In a previous report, John A. Rogers' team demonstrated the feasibility of wrapping a flexible sensor array onto a cardiac balloon catheter to measure physiological signals[39]. In OSA obstruction location detection, not only does the surrounding pressure value need to be detected in situ at the obstruction location, but the pressure distribution should also be considered when locating the obstruction. Currently, flexible sensor arrays are usually first fabricated on a flat surface and then transferred to the catheter surface using a wrapping or imprinting technique, which may cause additional strain when the sensor dimension mismatches the curvature and length[37,47,48]. Progress in femtosecond laser development enabled the engraving of a 3D sensing microstructure on the catheter wall, resulting in a high-fidelity and reliable structure for circumferential sensing in a long-ranged region[36,49,50]. The combination of these technologies and techniques could provide routes and tools that overcome the limitations and challenges in current sleep monitoring for OSA diagnosis.

In this study, we introduce a flexible, catheter-based pressure sensing array that can continuously monitor pressure distribution of the upper airway's soft tissue and provide high-resolution obstruction localization during sleep to assist OSA diagnosis and management in clinical practice (Fig. 1a). The sensing array reported in this study employs a flexible polyurethane (PU) gastric catheter as its substrate, with microstructures fabricated on the catheter surface using 3D femtosecond laser processing. A conductive layer of Poly(3,4-ethylenedioxythiophene)-poly(styrenesulfonate) (PEDOT: PSS) is applied to these microstructures, and a polydimethylsiloxane (PDMS) film coated with PEDOT: PSS wrap around the microstructured surface, creating an effective pressure-resistive sensor. Ten sensing units compose the sensing array, spanning its sensing area to approximately 6 cm in total, with each sensing unit measuring 5 mm in length. In vitro test results demonstrate a sensitivity of 38.1 Ω/mmHg, a response time of 660 ms, and excellent stability over 200 cycles, ensuring precise and timely pressure measurements. The application of this sensing array in the reversible OSA pig model confirms the effective detection of obstruction events, accurate localization, and local pressure measurement. The detected obstruction events correspond with the PSG

data, and the obstruction localization is consistent with the pig's upper airway stenosis CT imaging. These findings highlight the exceptional performance of this sensing array in accurately locating upper airway obstructions with high special resolution. Furthermore, its applicability extends beyond OSA, paving the way for innovative applications in traditional medical devices for pressure measurement in various human body cavities.

## Results

### Design of the flexible catheter-based sensor array

Figure 1 provides an overview and application scenarios for the catheter-based flexible pressure sensor array. While measuring OSA pressure, the catheter-based flexible pressure sensor array is inserted nasally into the upper airway, traversing the nasopharynx and oropharynx to the epiglottis (Fig. 1a). The sensor array uses a soft medical polyurethane gastric catheter (diameter of 2 mm) as the substrate, improving the overall flexibility and biocompatibility. The designed sensing area is approximately 6 cm long with a diameter of 2.5 mm, comprising 10 sensing units, each 5 mm long with a 1 mm inter-unit gap distance. The sensor array measures the temporal changes in pressure distribution during sleep, and the sensing area spans common obstruction sites such as the soft palate and posterior tongue region, facilitating high-resolution, precise positioning and absolute pressure measurement[1,51,52]. The flexible piezoresistive sensors were directly fabricated on the catheter's outer side via femtosecond laser 3D engraving functional microstructures on the curved surface. This approach minimizes the additional stress induced by circumventing conventional planar fabrication and device transfer processes to ensure high-fidelity signal detection. The sensor array is wrapped by a medical-grade polyurethane waterproof film and encapsulated with PDMS for biocompatibility.

The sensor consists of a four-layer structure. The conductive layer is attached to the surface of the catheter wall microstructure, and the elastic membrane substrate, featuring the conductive layer, is wrapped around the microstructure, as depicted in Fig. 1b. The photo of the flexible catheter-based pressure sensing array is shown in Fig. 1c. According to the cross-sectional schematic of a single sensing unit in Fig. 1d, the elastic membrane substrate wraps around the microstructured catheter, generating contact between the inner conductive layer of the elastic membrane and the outer conductive layer on the catheter's surface. Electrodes are positioned at both axial ends of the sensor, forming a circular ring around the catheter's surface. For stable electrical signal connection, one end of an enameled wire is bonded to the circular ring electrode on the catheter's external surface using an elastic conductive adhesive. The other end threads through the catheter wall, wiring inside the catheter and transmitting the electrical signal to the catheter end. Excellent flexibility and small diameter enable the sensor array to adapt to the angle of the upper airway, reducing motion artifacts as well as foreign body sensations during measurement.

### Sensor operational principle

The axial cross-section and simplified electrical model of the flexible piezoresistive sensor are illustrated in Fig. 2a. Each sensing unit's resistance ($R$) comprises three components: the conductive layer's resistance on the membrane ($R_{F1}$), the conductive layer's resistance on the flexible catheter base microstructure ($R_{F2}$), and the conductive layers' contact resistance ($R_C$). When the device is exposed to external contact pressure, the membrane deforms towards the catheter, changing the contact area ($S_C$) between the conductive layers, which leads to the variation of $R_C$ according to the law of resistance. As a result, the sensor's resistance ($R$) is altered. The relationship between $R$ and $S_C$ can be simplified into the formula $R = R_{F1} + a/(S_C + b)$, where $a$, $b$, and $R_{F1}$ are constants, and $S_C$ is the contact area between the conductive layers on the membrane and the microstructures. A more

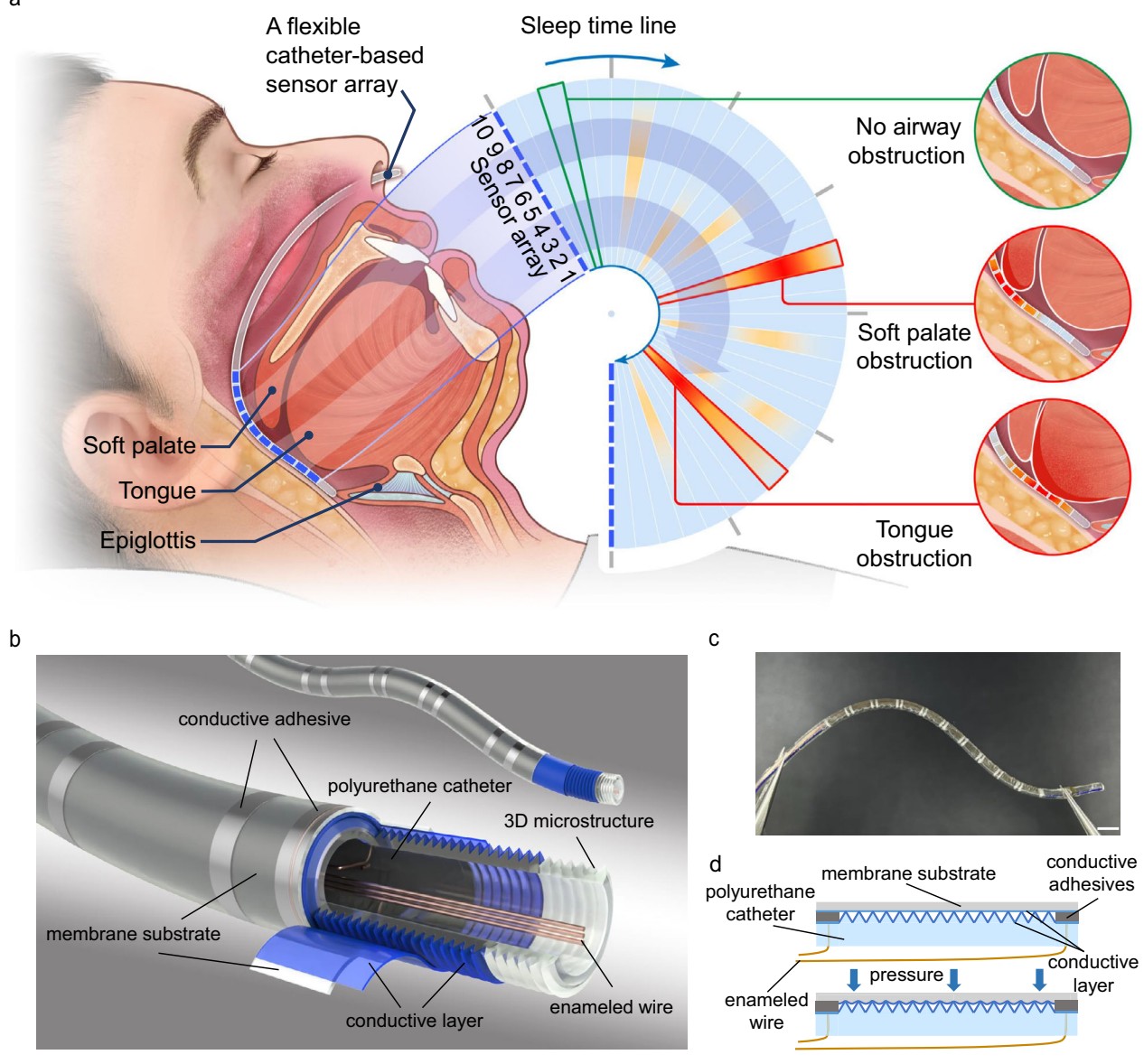

**Fig. 1 | A flexible catheter-based sensor array for monitoring pressure on upper airway soft tissues. a** Scenario illustration of the flexible catheter-based sensor array for obstructive sleep apnea (OSA) monitoring. A flexible catheter carrying the sensor array is designed to be inserted from the nasal cavity into the upper airway. The sensing array covers common upper airway obstruction sites such as the soft palate and posterior tongue region. **b** Schematic illustration of the flexible sensor array with a multilayer structure. **c** Photograph of the flexible catheter-based sensing array. Scale bar = 5 mm. **d** Schematic cross-section of a single sensing unit without pressure (top) and under pressure (bottom).

detailed derivation can be found in the Supplementary Note S1. After establishing the electrical model, an elastomeric contact deformation model is built and simulated using the finite element method. The relationship between $S_C$ and the applied pressure is derived by elaborating on the mechanical properties of the piezoresistive sensor with microstructures under pressure (Fig. 2b). Combining the mechanical property simulation and the simplified electrical model, a linear relationship between the external pressure and sensor resistance change $\Delta R$ can be found and shown in Fig. 2c.

## Characterization of the sensor array

Material selection, fabrication, and device performance evaluation are critical steps in obtaining a functional catheter-based flexible sensor. PDMS was chosen as the flexible sensor membrane material due to its elasticity and biocompatibility[53,54]. Conductive polymer was coated on the PDMS membrane and catheter microstructure surfaces. The conductive polymer contains three components: PEDOT: PSS, polyvinyl alcohol (PVA), and ethylene glycol (EG). Studies have shown that a higher mass fraction of doped PVA will improve the tensile limit of the conductive layer but reduce the electrical conductivity of the conductive layer[55,56]. Adding EG can compensate for the conductivity losses to ensure proper resistance for sensing purposes[57,58]. After conducting tests and comparing resistance levels of conductive polymer films with different PVA and EG doping ratios, we selected a doping ratio of 65 wt% PVA and 5.3 wt% EG in the PEDOT: PSS solution, achieving a balance of tensile properties and conductivity. Detailed material characteristics for different doping ratios and material selection processes can be found in Supplementary Fig. 1.

A femtosecond laser with a four-axis displacement fabrication platform etched the triangular cross-sectional microstructures on the catheter wall (Fig. 2d). The uniform microstructure can be seen under the scanning electron microscope (Fig. 2e). For device calibration and

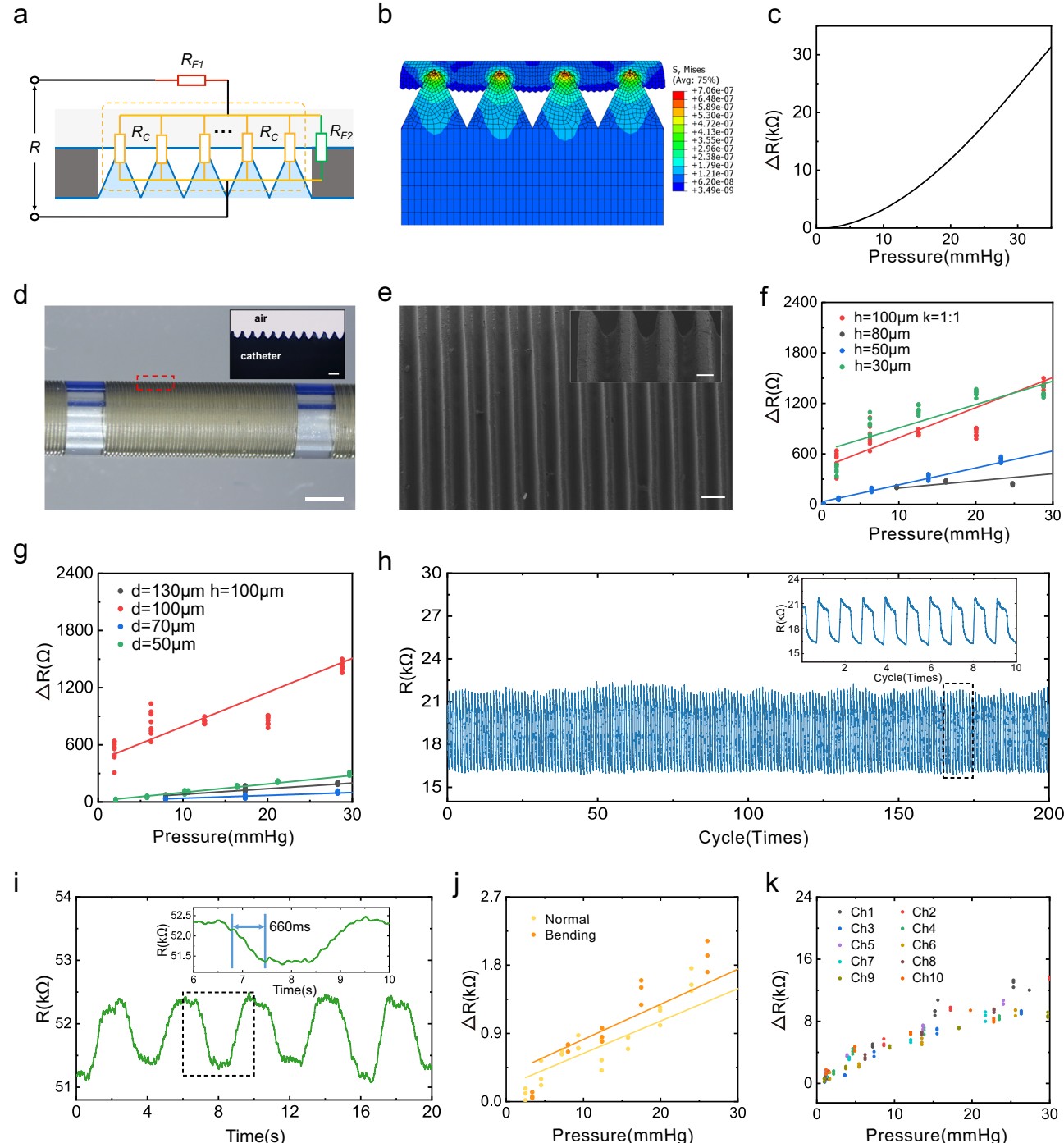

**Fig. 2 | Operational principle and characterization of the sensor. a** The simplified electrical model of the flexible piezoresistive sensor. **b** Simulation results of piezoresistive sensors' microstructural elastomeric contact deformation reveal the relationship between $S_C$ and applied pressure. **c** The relationship between the external pressure and sensor resistance change ($\Delta R$). **d** Photograph of the catheter after the microstructure has been etched on the catheter wall by femtosecond laser. Scale bar = 5 mm. Inset: enlarged view of the triangular cross-sectional microstructures on the catheter wall under a metallurgical microscope. Scale bar = 100 μm. **e** SEM image of uniformly distributed microstructures etched on the catheter surface by femtosecond laser. Scale bar = 100 μm. Inset: magnified side-view SEM image of the microstructures. Scale bar = 100 μm. **f** Resistance change ($\Delta R$) versus different pressures for sensors with the aspect ratio $K = 1:1$ at various depths (100, 80, 50, and 30 μm). **g** Resistance change ($\Delta R$) versus different pressures for sensors with the same depth of 100 μm at different widths (130, 100, 70, and 50 μm). **h** Relative change in sensor resistance response during 200 cycles of compression testing. **i** Response time of the sensor when 30 mmHg pressure is applied. **j** Resistance change ($\Delta R$) versus different pressures for sensors in the natural state and the 10 cm diameter bent state. **k** Resistance responses of each sensor unit in the same sensor array over the target pressure range.

performance evaluation, an in vitro model testing platform has been established. A periodic pressure was applied on the sensor via the motor, with real-time sensor pressure signals read out by the circuit and displayed and stored on the graphic user interface (GUI). The

sensitivity test within the target pressure measurement range was performed on sensors with identical aspect ratios but varying depths h of the microstructure. Microscopic images of the microstructure with different parameters are shown in Supplementary Fig. 2. The pressure

change induced resistance variation of the sensor with the same aspect ratio of $K = 1{:}1$ but different depths (100, 80, 50, and 30 μm) is shown in Fig. 2f. Among the four sensor depth parameters, $h = 100$ μm demonstrates superior sensitivity with 38.1 Ω/mmHg, while sensitivities for depths of 80 μm and 50 μm are measured as 8.6 Ω/mmHg and 20.1 Ω/mmHg respectively. The sensor with $h = 30$ μm was close to saturation at pressures greater than 20 mmHg and did not reach our target pressure (30 mmHg), which may be due to the smaller size of the microstructure reaching saturation during deformation. The sensitivity of four sensor groups with a microstructure depth $h$ of 100 μm and varying widths d was subsequently evaluated, as shown in Fig. 2g. All four sets of sensors exhibited linear responses within the desired measurement range of 0–30 mmHg. The sensor sensitivities at depths of 50, 70, 100, and 130 μm were determined to be 9.0, 3.8, 38.1, and 6.3 Ω/mmHg respectively. Through the above test results, the 1:1 aspect ratio, 100 μm depth sensor exhibited the highest sensitivity of 38.1 Ω/mmHg and was thus selected for further in vitro model testing and animal experiments.

To evaluate the sensor's response time, 0.25 Hz periodical stimuli were applied to the sensing unit using a piezoelectric ceramic, measuring the sensor's response time of 660 ms (Fig. 2i). This response time is regarded as adequate for monitoring sleep apnea, considering that the standard human respiration rate is between 12–20 breaths per minute. Repeatability experiments were conducted to verify the sensor's stability after repeated compressions, which is crucial for long-term sleep monitoring. In a 200-cycle compression test, the sensor's resistance response remained consistent, with less than 6% change in amplitude from the experiment's start (Fig. 2h). Bending and waterproofing tests were conducted to evaluate sensor performance in humid and curved upper airway environments. The bending tests show an 8.6% sensitivity reduction (Fig. 2j), attributed to increased internal stress in the membrane that hindered the pressure-induced deformation. Following the waterproof test, a decrease in sensor sensitivity by 20.1% was observed (Supplementary Fig. 3), attributed to the infiltration of water vapor through the PDMS at the anterior opening of the catheter. The resistance responses among the sensor units exhibited excellent consistency within the target pressure range, as illustrated in Fig. 2k. To investigate the comfort and flexibility of the sensor array, we collected several clinical catheters currently used to enter the upper airway or esophagus via the nasal cavity. The tensile and bending moduli of the sensor arrays and other tubular medical catheters were tested (Supplementary Note S2), with the comparative results presented in Supplementary Table 1. Our sensor array exhibits a bending modulus of 5.86 MPa and a tensile modulus of 26.13 MPa. These values are comparable to those of other tubular catheters, suggesting that the sensor array offers similar comfort during use as nasal catheters adapted for clinical settings.

## Animal experiment

A reversible and safe OSA model was established in four Bama pigs by palatal-pharyngeal injection of a cross-linked sodium hyaluronate mixture (Supplementary Note S3), with one pig as the control group and three pigs as the experimental group[59]. PSG, upper airway CT scan, and catheter-based pressure sensor array monitoring were performed before modeling, on the day of modeling, and 2, 7, and 14 days after modeling. Under sedation-induced sleep, data were collected using the catheter-based pressure sensor array to monitor obstructed positions and pressures in the upper airway with PSG (Supplementary Fig. 4). The sensor array, with sensors numbered from Ch1 to Ch10, is applied to the experimental group covering the range from epiglottis to soft palate. The catheter carrying the sensor array stopped at the epiglottis, as shown in Supplementary Fig. 5. The application scenario of the catheter-based flexible pressure sensor array on the OSA model pig and the corresponding CT sagittal image of the OSA model pig is depicted in Fig. 3a. The control group underwent PSG monitoring alone, without sensor array placement. The sensor array readout was compared and analyzed against the CT and PSG data to evaluate the sensor array's effectiveness in identifying OSA obstruction locations and pressures.

## Response of the catheter-based sensor array to respiratory events

The PSG analysis software combined with the video recordings determined the reality of the OSA event, which facilitated the comparison of the signals of the OSA event. The sensor response matched the PSG signal during the OSA event and accurately reflected the airway status, demonstrating the sensor array's precise responsiveness to OSA events in model animals. In the experimental group, a representative OSA event among three OSA model pigs is shown in Fig. 3b and Supplementary Movie S1. Figure 3b shows the PSG measurement and the corresponding sensing array response between 90 and 120 min in the trail, highlighting the areas where PSG-reported OSA events occurred. An OSA event happened where the upper airway was obstructed at 116.7 min, and the PSG thorax and abdominal signals displayed an increase in breathing movement and breathing effort due to long-term hypoxia in the OSA model pigs. As the time of airway obstruction increases, the blood oxygen level of the OSA model pigs gradually decreases. Correspondingly, the sensor array demonstrated a decrease in the resistance of Ch2, Ch3, and Ch5, indicating an increase in the contact pressure at the obstructed position. At 117.5 min, the PSG thorax and abdominal signals showed significant amplitude waves, indicating that the pig breathed deeply after the airway was opened, representing the cessation of the OSA event. Over the next 0.5 min, the PSG thorax and abdominal signals gradually returned to the pre-OSA event level and oxygen saturation gradually recovered. At the same time, the sensor array records matching fluctuations in Ch2, Ch3, and Ch5 to correspond to these events. The obstruction-induced airway collapse generates a static contact pressure during 116.8–117.2 min, and as the OSA event terminates and the airway opens again, the sensor resistance increases rapidly, indicating the pressure released in the airway. By analyzing the sensor array data compared to the video, we can also identify additional events occurring around the pharynx, such as snoring (Supplementary Fig. 6), chewing and swallowing (Supplementary Fig. 7). These activities exhibited distinct signal characteristics that differentiate them from the obstruction events. This capability underscores the sensor array's effectiveness in distinguishing various respiratory-related behaviors in the upper airway.

## Localization of obstruction position by catheter-based sensor array

The capability of the pressure sensor array to accurately determine the position of obstructions was investigated by comparing the position of the stenosis area in the CT image with the pressure response measured by the sensor array. Figure 3c is the resulting representative plot among the experimental group pigs. The location of the minimum cross-sectional area in the upper airway CT reconstruction model was compared to the pressure distribution heatmap generated by the sensor array at different time points, with the epiglottis serving as the reference plane.

The injection of a sodium hyaluronate mixture into the soft palate and lateral pharyngeal wall of Bama pigs during the establishment of an OSA animal model resulted in a change in the distance between the epiglottis and the location of the minimum upper airway cross-section from 17.46 mm to 19.06 mm. On the day of modeling, CT imaging revealed a distinct stenotic within the airway (Fig. 3a, c). The heat map generated by the sensor array on the day of modeling showed high contact pressure values at Ch2, with local pressures exceeding 25 mmHg, while the pressures at the other sensing units remained below 10 mmHg. Depending on the distribution of the sensor units, the

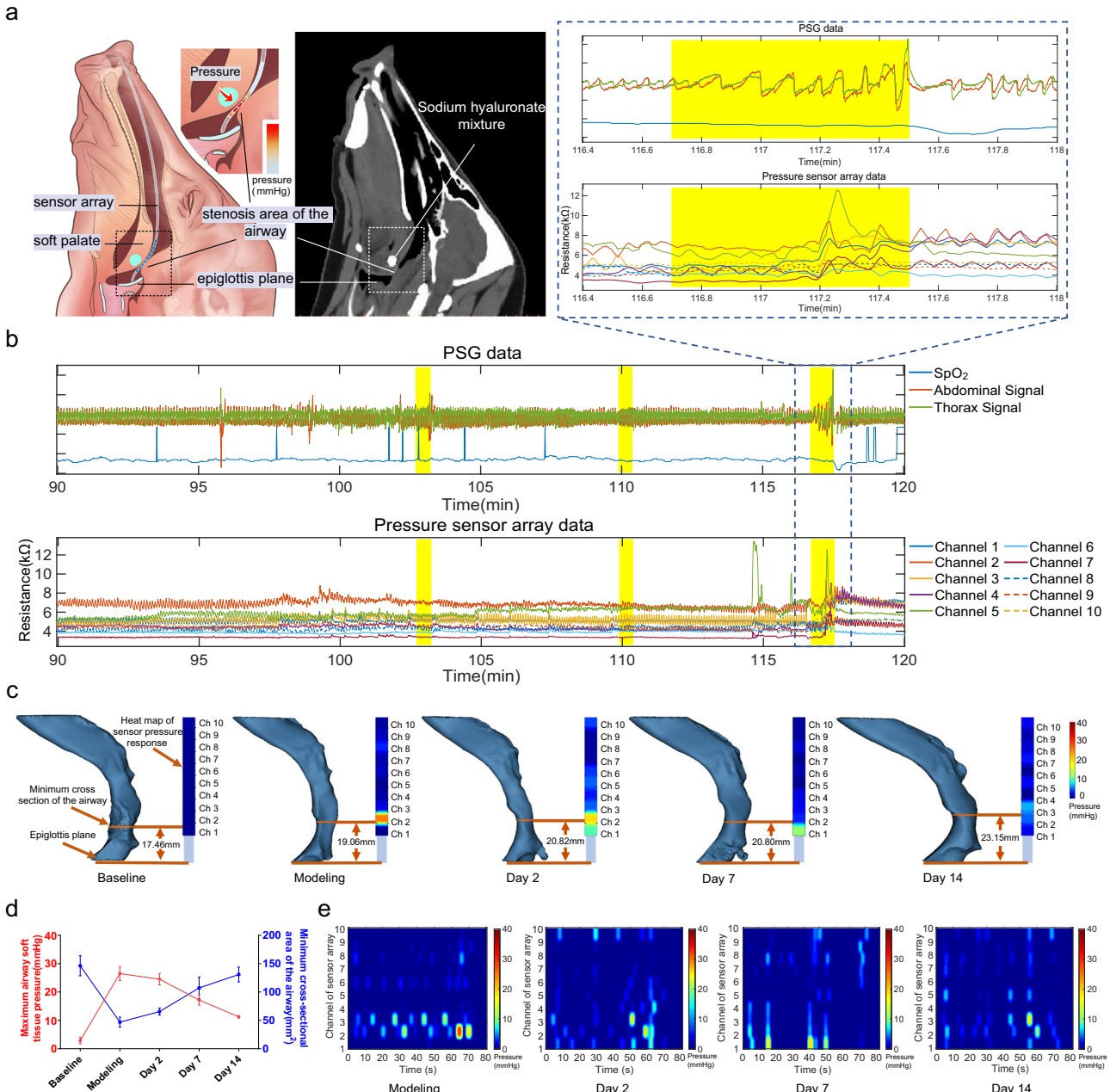

**Fig. 3 | In vivo demonstration of a flexible catheter-based pressure sensing array in the OSA pig model. a** Scenario illustration of the flexible catheter-based sensor array placed inside the upper airway of OSA model pigs for soft tissue pressure monitoring (left) and the corresponding upper airway CT image of OSA model pigs (right). **b** PSG measurements, including thoracic motion signals, abdominal motion signals, and blood oxygen saturation, and the corresponding sensing array responses during 90 to 120 min of sleep monitoring, with PSG-reported OSA events highlighted. Inset: zoom-in illustration of a representative OSA event region. **c** The location of the minimum cross-sectional area in the upper airway CT images at different time points before and after modeling and the pressure response measured by the sensor array during the at the time of the OSA event. **d** The minimum cross-sectional area of the upper airway and the maximum airway soft tissue pressure during OSA events in OSA model pigs at different time points before and after modeling. $n = 3$ independent samples, data are presented as mean values +/− SEM. **e** A set of heat maps of obstruction pressure distribution during OSA events at different time points.

sensing range of Ch2 is 18 to 24 mm from the tip of the catheter. The minimum cross-section of the airway was within the sensing range of Ch2. On the second day after modeling, the CT image showed a distance of 20.82 mm between the minimum cross-section of the airway and the epiglottis. Heat maps of the pressures generated by the sensor array showed that the pressures at Ch1 and Ch2 were over 20 mm Hg, while the pressures at the other sensors were less than 10 mmHg. The sensing areas of Ch1 and Ch2 were distributed within a range of 12–24 mm from the tip of the catheter. On day 7 after modeling, the position of the airway obstruction moved to Ch1. CT imaging showed

that the distance between the minimum cross-section of the airway and the epiglottis was 20.80 mm, which is very close to the sensing range of Ch1, in the range of 12–18 mm from the tip of the catheter. On day 14 after modeling, CT imaging showed that the distance between the minimum cross-section of the airway and the epiglottis was 23.15 mm, which was consistent with the pressure heat maps generated by the sensor arrays (Ch2 and Ch3). The same consistent results were observed in two other OSA model pigs (Supplementary Fig. 8). These measurement results confirmed that the locations observed by CT imaging corresponded with pressure distribution depicted by the heat

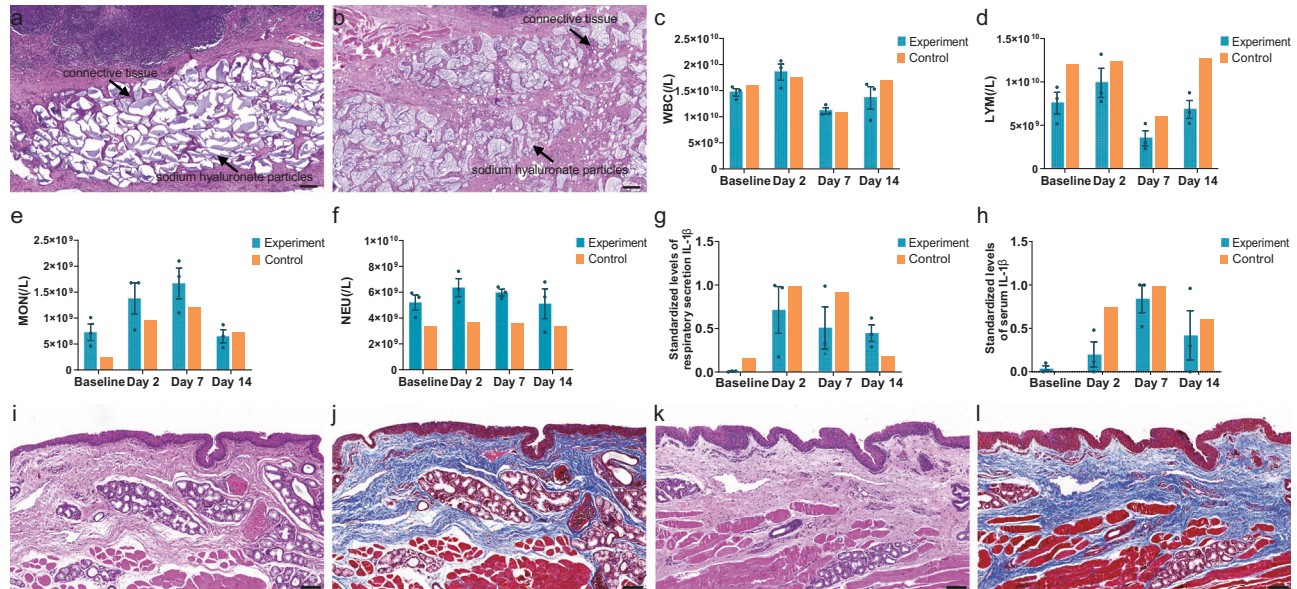

**Fig. 4 | The biocompatibility studies of the sensor.** Hematoxylin-eosin staining images of the injection area of OSA model pigs on day 14 (**a**) and day 28 (**b**) after modeling. Scale bar = 200 μm. **c**–**f** Serum levels of white blood cells (**c**), lymphocytes (**d**), neutrophils (**e**), and monocytes (**f**) in the experimental (*n* = 3) and control (*n* = 1) groups at different time points after modeling. Data are presented as mean values +/− SEM. **g**, **h** Levels of interleukin-1β (IL-1β) in upper airway secretions (**g**) and serum (**h**) of the experimental (*n* = 3) and control (*n* = 1) groups at different time points after modeling. Data are presented as mean values +/− SEM. Microscopic images of tissue from the posterior pharyngeal wall stained with hematoxylin-eosin (**i**) and Masson (**j**) after placing the sensing array in the upper airway of a normal pig for 24 h. Scale bar = 100 μm. Hematoxylin-eosin staining (**k**) and Masson staining (**l**) of tissues of the posterior pharyngeal wall of blank control pigs. Scale bar = 100 μm.

map generated by the sensor array, thus validating its precise ability to locate airway obstructions. In addition, the sensor array has the ability to accurately detect multiple site obstructions simultaneously in the airway (Supplementary Fig. 9).

Figure 3d shows the changing trends of the minimum cross-sectional area of the airway and the maximum pressure of the airway soft tissue during OSA events at each time point before and after modeling in the experimental group. Before modeling, the minimum cross-sectional area of the upper airway of all pigs was 146.21 ± 30.63 mm². With the establishment of the OSA animal model, the minimum cross-sectional area of the upper airway decreased to 46.88 ± 15.33 mm². The airway soft tissue pressure measured by the pressure sensing array showed an increase from 3.89 ± 1.98 mmHg to 26.49 ± 4.27 mmHg, consistent with the changes in upper airway CT. Over time, the sodium hyaluronate mixture in the injection area of the OSA model pig was gradually absorbed. CT showed that the minimum cross-sectional area of the upper airway gradually increased, and the airway soft tissue pressure also showed a consistent decreasing trend. On day 14 after modeling, the minimum cross-sectional area of the upper airway increased to 130.83 ± 44.06 mm², and the airway soft tissue pressure gradually decreased to 11.25 ± 0.76 mmHg. The respective data of the three OSA model pigs in the experimental group at each time point of the minimum cross-sectional area and maximum pressure of the airway are shown in Supplementary Fig. 10. Figure 3e, representative of the experimental group, shows a set of heat maps showing changes in the airway soft tissue pressure distribution over time at different time points. Heat maps of the airway soft tissue pressure distribution over time for each pig in the experimental group at different time points are displayed in Supplementary Fig. 11. These results demonstrate the potential of pressure sensing arrays for long-term, continuous upper airway soft tissue pressure monitoring.

## Biocompatibility assessment
The biocompatibility of the catheter-based flexible pressure sensor array has also been assessed in this study to ensure the safety of the

in vivo testing. Firstly, a histopathological examination of the injected area was performed to exclude the influence of the sodium hyaluronate mixture on the experimental results. On day 14 after modeling, the histopathological examination revealed the presence of aggregated sodium hyaluronate-filled particles within the lamina propria and newly formed connective tissue between these particles (Fig. 4a). On day 28 after modeling, there was an increase in the gaps between sodium hyaluronate particles within the lamina propria compared with day 14 after modeling, as well as an enhancement in the formation of connective tissue between the particles (Fig. 4b and Supplementary Fig. 12). These observations suggest that the injected sodium hyaluronate mixture was gradually absorbed over time, consistent with the gradual increase in minimum airway cross-sectional area and decrease in pressure shown by the sensor arrays. No cell mutations or inflammatory cell aggregation were observed at either time point. Serum analysis was performed within 14 days after modeling to assess representative inflammatory cells in both animal groups. In both experimental and control groups, white blood cells (WBC) (Fig. 4c) and lymphocytes (LYM) (Fig. 4d) increased on the second day after modeling, decreased on day 7, and returned to pre-modeling levels on day 14. Neutrophils (NEU) in both groups did not change significantly within 14 days after modeling (Fig. 4e). Monocytes (MON), indicating later-stage inflammation, also showed a similar pattern of gradual increase within 7 days, returning to pre-modeling levels on day 14 in both groups (Fig. 4f). Interleukin-1β (IL-1β) levels in upper airway secretions and serum were also tested to assess the inflammatory response. IL-1β in secretions from both groups showed the same trend, increasing on the second day after modeling and gradually decreasing afterward. (Fig. 4g). The serum IL-1β levels of both groups gradually increased within 7 days after modeling and then decreased on day 14. (Fig. 4h). This pattern of change correlated with the severity of OSA in the model animals. With the absorption of injected sodium hyaluronate, the severity of airway obstruction is reduced, and Apnea Hyponea− Index (AHI) events during sleep are

decreased (Supplementary Fig. 13), which leads to a gradual reduction in systemic and airway inflammation.

The biocompatibility for the catheter-based flexible pressure sensor array was further validated by being placed in the pig's upper airway for 24 h. Histopathological examination of the tissue of the posterior pharyngeal wall in contact with the sensor array showed a clear demarcation and layering of the mucosa, lamina propria, and muscular (Fig. 4i). Abnormal arrangement and thickening of fibrous connective tissue were not observed in this region (Fig. 4j). These microscopic observations were consistent with those of the untreated control pig (Fig. 4k, l), and the tissue collagen fiber content was similar between the two groups (Supplementary Fig. 14). These comprehensive tests on serum and respiratory tissue demonstrated that the in vivo placement of the sensor array in this experiment did not elicit any additional systemic or local inflammatory responses or tissue abnormalities, thereby confirming its excellent biocompatibility.

## Discussion

Several techniques for OSA obstruction site localization have been proposed in the literature, including multi-sensor catheters[28–30,60,61], optical coherence tomography (OCT)[62–64], flow pattern analysis[65], local ultrasound[66–69], and snoring sound frequency analysis[70–77]. While these methods conducted promising attempts from various technical routes, certain limitations still hindered their performances. Multi-sensor catheters that enter the esophagus through the upper airway primarily detect changes in air pressure in different parts of the respiratory process, rather than measuring soft tissue contact pressure directly. Multi-sensor catheters with FBG technology can measure airway contact pressure gradients accurately, but their protective coating and cable jacket may affect flexibility and patient comfort. OCT provides real-time 3D airway imaging but is limited by respiratory movement interference and high cost. Local ultrasound is effective for real-time soft tissue monitoring but struggles with deeper structure measurement and high operator skill dependency. Non-invasive flow patterns and snoring sound analysis are indirect measurements with low accuracy and spatial resolution, particularly in multiple obstruction site identification. Snoring sound analysis is also easily affected by external factors like background noise and patient positioning.

This study presents a catheter-based flexible pressure sensor array that provides continuous monitoring of soft tissue contact pressure in the upper airway, enabling high-resolution obstruction localization. The array is designed to facilitate the diagnosis and treatment of OSA, addressing the existing challenges and limitations of conventional sleep monitoring tools and clinical practices (Supplementary Table 2). By providing direct, in-situ pressure distribution data along the airway, the sensor offers valuable insights for effective OSA management. A versatile 3D femtosecond laser fabrication process allows for adapting the sensor's design across various materials and applications, resulting in a flexible, reliable, and user-friendly system. The sensor's mechanical properties, with a bending modulus of 5.86 MPa and a tensile modulus of 26.13 MPa, ensure sufficient flexibility to minimize patient discomfort. In vitro and in vivo validation using a reversible OSA pig model demonstrated the array's capacity to detect and localize airway obstructions precisely, consistent with polysomnography and CT imaging. The array's ability to monitor distributed pressure and detect millimeter-level obstructions, including simultaneous multiple sites, underscores its potential for accurate OSA diagnosis.

Despite the advantages of this catheter-based sensor array, the device can be further optimized and tested for future clinical human trials. The reduction of the device size and the increase in comfort could be one of the directions. Additionally, although the sensor performed well in controlled animal models, differences between OSA patients and animals present a challenge for practical clinical use.

This study proved the potential of the catheter-based sensor array for long-term, continuous upper airway pressure assessment, coupled with its demonstrated biocompatibility, paving the way for further human-based research. The findings are particularly significant for OSA diagnosis and treatment planning in clinical settings, especially in determining surgical intervention sites and extents. Moreover, the scope of its applicability goes beyond OSA and extends to conventional medical devices employed for measuring pressure in various human body cavities. It has the potential to revolutionize the medical field by introducing innovative applications in diverse human body cavities.

## Methods

### Fabrication of microstructures using a femtosecond laser
Microstructures were fabricated on the surface of the catheter using a femtosecond laser system (Phidia-c, Uptek, NY, US), with an output power of 4.0 W and a pulse duration of 120 fs. The system comprised a four-axis displacement platform with mutually perpendicular $X$, $Y$, and $Z$ axes, along with a rotational $U$ axis (Supplementary Fig. 15). By focusing the femtosecond laser close to the catheter surface, a 360° rotation of the $U$ axis is achieved within a single engraving cycle. An array of microstructures featuring triangular cross-sections can be produced on the catheter by adjusting the stepping distance along the $X$-axis (Supplementary Fig. 16). Morphology analysis, including width-to-depth ratio and depth measurements of these microstructures, was conducted during fabrication using a metallographic microscope (UM200i, UOP, China) and further characterized via Scanning Electron Microscope imaging (GeminiSEM 500, Zeiss, German).

### Fabrication of PDMS conductive films
The PDMS base was mixed with an agent (SYLGARD 184, Dow Corning, USA) in a 10:1 mass ratio and thoroughly homogenized. A vacuum pump was used to remove air bubbles from the mixture. The PDMS mixture was spin-coated on a 70 mm × 70 mm glass slide (2000 rpm, 30 s) and then cured at 120 °C for 10 min to obtain the PDMS film substrate.

### Fabrication of conductive layers
Conductive layers were prepared by doping PEDOT: PSS solution (PH1000, Clevios, Germany) with 65 wt% PVA (Polyvinyl Alcohol 1799, Anjie Chemistry, China) and 5.3 wt% EG (Anjie Chemistry, China) at 40 °C for 30 min at 150 r/min, forming a conductive mixture. The PDMS films and catheters were subjected to surface treatment to improve their hydrophilicity by plasma. The conductive mixture was spin-coated (2000 rpm, 30 s) and dried at 120 °C for 15 min. Subsequently, the coated layer was further post-treated with EG and dried again at the same temperature to create the sensor's conductive layer. To electrically isolate individual units in the sensor array and prevent electrical crosstalk, tape was applied to the smooth surfaces of the catheters between units before applying the conductive layer. The tape was removed after the EG post-treatment, resulting in discontinuous conductive layers on the catheters.

### Fabrication of sensor array
The insulation was removed from one end of a 70 μm diameter enameled wire, which was then wrapped around the start of the microstructured unit on the catheter surface. The other end was threaded through the catheter wall into the lumen and led out at the distal opening of the catheter. Conductive silver adhesive (AS7126, SHAREX, China) was applied around the exposed wire on the catheter surface and thermally dried at 120 °C for 30 min on a hotplate, forming a catheter-based microstructure array with electrodes (Supplementary Fig. 17). The PDMS conductive film was wrapped around the catheter surface, aligning it with the microstructured conductive layer to form a four-layer structure consisting of two elastomers sandwiching two

conductive layers, facilitated by PU-based medical waterproof tape. The PDMS base, curing agent, and hexane were mixed in a 10:1:5 mass ratio and uniformly applied to the surface of the sensor array. Subsequently, the device was packaged on a heating plate at 120 °C until complete curing. The detailed manufacturing process of the sensor array is shown in Supplementary Fig. 18.

## Construction of the sensor mechanical model

To validate the deformation and mechanical characteristics of a piezoresistive sensor with microstructures under pressure, we established a mechanical model of elastomer contact deformation in commercial software ABAQUS 2022 (Dassault Systèmes, Vélizy-Villacoublay, France) based on the sensor design. The overall mechanical deformation of the sensor can be approximated by considering the local microstructure-film deformation on the two-dimensional section, given that the microstructure on the catheter is symmetrically distributed and repeats periodically along the axial direction. This allowed us to explore the relationship between the contact area of conductive layers on both sides and applied pressure. When the sensor is subjected to external pressure, the direction of force application is from the outer to the inner of the catheter. In modeling a local cross-section of the sensor, the load can be simplified as a pressure applied vertically downward from the film surface. The PDMS film thickness was set to 40 μm, while the overall height of the polygon matched the guide catheter wall thickness of 500 μm. Additionally, an isosceles triangle with a base width of 100 μm was established at a height of 100 μm. The Young's modulus of the PDMS film material was set at 2 MPa, with a Poisson's ratio of 0.495 and a density of 1.028 g/cm³, while the polyurethane catheter material with surface microstructures demonstrated a Young's modulus, Poisson's ratio, and density of 10 MPa, 0.470, and 1.260 g/cm³ respectively.

## Signal acquisition and analysis

Signal acquisition for the sensor array was conducted using a wireless, portable acquisition system. The system comprised two main components: a hardware circuit board and PC-based acquisition software. The hardware board featured an interface circuit, a microcontroller chip (STM32F722RET6), and a Bluetooth module powered by a 5 V portable charger (Supplementary Fig. 19). The interface circuit depicted in Supplementary Fig. 20 facilitates the conversion of the device's electrical response into electronic signals, which are subsequently amplified and directed to the microcontroller for integrated processing. The hardware sampling rate is set at 83 Hz. A pressure sensor was used for real-time acquisition of pressure signals, with the system enabling real-time observation and storage of sensor responses. The data were later exported for signal processing and analysis using MATLAB (R2020b).

## Device calibration and performance evaluation via vitro testing platform

An in vitro test platform was established, referencing Young's modulus of the upper respiratory tract's surrounding tissues (Supplementary Fig. 21). The sensitivity of sensors with varying microstructure parameters was evaluated using the test platform and data acquisition system. The platform consisted of four parts: a pressure application and release device, a pressure transmission device, a pressure acquisition device, and silicone rubber pads. The pressure application and release device comprised a push rod, a motor drive circuit, and optical fixtures. The Arduino was used for programmable control of the direction and speed of the push rod motor (STA50, Bolin Intelligent Technology, China), providing periodic pressure changes to the sensor through the reciprocating motion of the push rod. Optical fixtures determined the relationship between the pressure direction and the position of the sensor array, enabling controlled pressure application. The pressure transmission device included an airbag and glass slides,

with the airbag placed between the push rod motor and the glass slides. A pressure sensor chip was connected to the airbag via a tube, and the pressure value inside the airbag was read and calculated. Silicone rubber pads made from Ecoflex material (Smooth-On, Inc., Macungie, PA, USA), with Young's modulus close to that of airway soft tissue (2.98 ± 0.44 kPa), were placed on either side of the sensor, and PU-based medical waterproof tape was used to encapsulate the pads, reducing stickiness. A periodic pressure was applied on the sensor via the motor, with real-time sensor pressure signals read out by the circuit and displayed and stored on the GUI. To assess the device's ability of waterproof performance, it was submerged for 8 h in water at 37 °C. The sensitivity of the sensor was measured before and after the immersion. Furthermore, we evaluated the sensor's sensing ability in a bent state by fastening the catheter carrying the sensor array with a bending radius of 10 cm on the test platform. The sensitivities from both the bent and non-bent states were compared.

## Response time testing

As shown in Supplementary Fig. 22, the experiment utilized a piezoelectric ceramics bimorph with a response time of 2.7 ms. A square wave signal output from a signal generator was amplified using a linear voltage amplifier and applied to the piezoelectric piece, driving the piezoelectric piece to deform at a set frequency. The sensor was placed under the end of the piezoelectric piece to monitor the response time of the sensor under periodic rapid pressure changes.

## Establishment of the OSA bama pig model

Four 6-month-old adult male Bama pigs (purchased from Tianjin Bainong Experimental Animal Breeding Technology Co., Ltd., Tianjin, China) were selected and adaptively fed until 32–35 kg for modeling, with detailed manufacturing steps in Supplementary Note S3. Bama pigs were injected with an appropriate amount of anesthetic to induce sleep. The syringe was filled with a mixture and injected at multiple points into the bilateral pharyngeal palatine area of the animals (Supplementary Fig. 23). Injection ceased when snoring sounds appeared and typical apnea patterns were observed on PSG. The successful criterion for OSA modeling was defined as cessation of oronasal airflow for at least 10 s, with a corresponding decrease in blood oxygen saturation of ≥ 4%.

## PSG signal detection and analysis

Sleep data were collected using a multi-lead polysomnography (Smote PSG, COMPUMEDICS, Australia). The collected data included electrocardiography (ECG), electroencephalography (EEG), oxygen saturation (SpO2), airflow, thoracic movement, abdominal movement, leg movement, and body position. The built-in analysis software (ProFusion PSG4) was used for automatic extraction and analysis of respiratory events.

## Device signal acquisition and processing in animal models

As shown in Supplementary Fig. 4, the same portable acquisition system was utilized for signal acquisition of the model animal sensing array and in vitro experiments. In MATLAB software, the level signals were converted to voltage signals, and the sensor resistance was calculated using the formula of interface negative feedback amplifier circuit. Median filtering was applied to remove spikes, followed by average filtering to eliminate high-frequency noise. To comprehensively analyze PSG and sensor signals, raw data from PSG signals, including thoracic and abdominal movements, blood oxygen levels, airflow rates, and electrocardiography, were extracted and imported into MATLAB. The PSG data and pressure sensor array signals were synchronized using timestamps. Based on the location of obstructive sleep apnea events reported by PSG analysis software and video recordings, false OSA events were eliminated to identify the corresponding regions of both signal sets during actual OSA events.

MATLAB was utilized to extract and normalize waveform amplitudes corresponding to each sensor unit in the sensor array signal, resulting in a heatmap illustrating the distribution patterns of the sensor array output signal.

## Device signal comparison with PSG and CT data
Four pigs were positioned in the same prone position for spiral CT scanning (NeuViz 16 Classic, Shenzhen Mindray Bio-Medical Electronics Co., Ltd., Shenzhen, China), maintaining consistent CT parameters (120 kV, 400 mAs) to acquire upper airway data. CT data at each time point were imported into Mimics Medical 21.0 software in DICOM format. A threshold of −1024 HU to −150 HU was selected to create a mask, identifying landmarks such as the mid-palatine suture, posterior edge of the hard palate, greater palatine foramen, foramen magnum and the lower edge of the first cervical vertebra to segment the upper airway from the greater palatine foramen to the lower edge of the first cervical vertebra. The software's built-in measurement tools were used to calculate the cross-sectional area of the respiratory tract at each CT layer. The narrowest cross-sectional area was selected and measured three times for analysis. The distance between the minimum cross-section of the airway and the epiglottis was measured at the sagittal view of CT, with the epiglottis plane serving as a reference. This measurement was then compared to the sensor array's obstruction position to assess its capability in pressure distribution and obstruction localization.

## Blood inflammatory cell examination
Venous blood samples were systematically collected from model animals at each designated time point. Using laser flow cytometry analysis (BC-2800 Vet, Mindray, China), white blood cells, neutrophils, monocytes, and lymphocytes were quantified in the venous blood.

## Biocompatibility pathological examination
Tissue biopsies were taken from the left soft palate on day 14 after modeling, with a biopsy sample size of 1 cm × 1.5 cm. Tissue from the posterior pharyngeal wall (1 cm × 1.5 cm) in contact with the sensing array was obtained 24 h after the sensor array was placed in the upper airway of blank Bama pigs. All specimens were fixed in 4% paraformaldehyde for 12 h and then dehydrated, clarified, paraffin-embedded, sectioned, and stained with HE and Masson. The staining results were scanned as electronic images and analyzed using CaseViewer 2.4 software. The image analysis software program "ImageJ" (version 1.53 g, National Institutes of Health, Bethesda, MD, USA) was used for quantitative comparative analysis of the tissue histology.

## Enzyme-linked immunosorbent assay (ELISA)
Venous blood collected from model pigs at each time point underwent centrifugation at $2327 \times g$ for 10 min, and serum was extracted. Throat swabs were employed to collect airway secretions from OSA pigs at each designated time point, which were then placed in 2 ml of physiological saline, mixed thoroughly, and the supernatant was extracted. IL-1β levels were measured using the IL-1β ELISA Kit (Solarbio, Beijing, China) according to the manufacturer's instructions. Absorbance measurements were performed using a BioTek Synergy H1 multi-mode microplate reader (BioTek Instruments, Winooski, VT, USA).

## Statistical analysis and reproducibility
At least three independent experiments of each type have been done and produced consistent results. All data were analyzed using the SPSS 21.0 statistical package (SPSS Inc., Chicago, IL, USA). For normally distributed and variance-aligned data, data were analyzed using independent samples $t$-test or paired $t$-test. Two-tailed tests were used for all tests. Differences of *$p < 0.05$ denote significance compared with other groups. GraphPad Prism 9.3 and Origin 2018b were used for graphing.

## Ethics
This experiment is approved by the Biomedical Ethics Committee of Beihang University, approval number BM20230259. Every experiment involving animals has been carried out following a protocol approved by the ethical commission.

## Reporting summary
Further information on research design is available in the Nature Portfolio Reporting Summary linked to this article.

## Data availability
All data supporting the findings of this study are available within the article and its supplementary files. Any additional requests for information can be directed to, and will be fulfilled by, the corresponding authors. Source data are provided with this paper.

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

## Acknowledgements

This study was supported by the National Key Research and Development Program of China 2023YFC3603500 (R.L.), the National Natural Science Foundation of China 12332019, U20A20390 and T2288101 (Y.F.), the Clinical Medicine Plus X-Young Scholars Project of Peking University PKU2024LCXQ028 and the Fundamental Research Funds for the Central Universities (S.Z.). We would like to extend our sincere gratitude to Dr. Haiyang Wang from Beihang University, and Dr. Xiaolong Sui and Dr. Tong Wang from the Pathology Department at Yantai Yuhuangding Hospital for their invaluable assistance with the preparation and interpretation of pathological slides.

## Author contributions

X.M. and J.S. were responsible for sensor fabrication, data acquisition, and analysis. X.M., Yan.F., and J.L. were responsible for 3D microstructure processing. J.S. and X.M. performed the animal experiments. T.J. and S.Z. supervised the animal experiments. J.S. performed the histological staining and ELISA analyses. X.M., P.Z., and Z.L. developed the MATLAB scripts for data processing. C.H. and J.W. assisted with some of the data acquisition. R.L. and Y.Fan. were responsible for the experimental design and supervision. J.S. and X.M. wrote and edited the manuscript. R.L., Y.Fan., and S.Z. revised and reviewed the manuscript. All authors participated in the study and critically reviewed the manuscript.

## Competing interests

The authors declare no competing interests.
