## [Transparent Peer Review file · Nature Communications]

A Flexible Catheter-Based Sensor Array for Upper Airway Soft Tissues Pressure Monitoring

Corresponding Author: Professor Ruya Li

Version 0:

Reviewer comments:

Reviewer #1

(Remarks to the Author)

The authors provide a comprehensive description of a novel technology for assessing upper airway mechanics in the context of sleep apnoea. They provide compelling data on the validity of the technique in identifying obstructive episodes and the site of collapse. This is a current gap in the field, and a solution would advance the principle of precision medicine by allowing selection of patients for targeted interventions based on the primary anatomical site of collaps.

Numerous previous attempts to assess the site of collapse using diverse techniques have been reported, and the authors should acknowledge such studies. These include studies of a multisensor catheter (doi.org/10.1093/sleep/29.5.666), optical coherence tomography (doi.org/10.1364/OE.11.001817), flow shape analysis (doi.org/10.1016/j.chest.2017.06.017), and snoring analysis ([10.1109/EMBC44109.2020.9175591](https://doi.org/10.1109/EMBC44109.2020.9175591)).

The clinical utility of such a technique in humans remains to be examined. However, given the invasive nature of the technique (requiring a catheter in the upper airway) it is difficult to see how this approach would be adopted in clinical practice. Nevertheless it could be a useful research tool.

One aspect that is relevant to human OSA is that there are usually multiple sites of airway collapse. The authors should address the question of whether their approach is able to detect multiple simultaneous sites of collapse. Or does complete collapse at one site prevent detection of collapse at other sites?

Reviewer #2

(Remarks to the Author)

The authors report a catheter-based flexible pressure sensor array that continuously monitors soft tissue pressure in the upper airway and facilitates at the millimeter level. The sensor is designed and tested to prove the potential of the catheter-based sensor array for long-term, continuous upper airway pressure assessment. The comments can be found below. The authors provide a long term solution for upper airway soft tissues pressure monitoring. How about the comfort and safety of this sensor? How about the flexibility of it?

Are there any other similar solutions for the soft tissues pressure monitoring?

Is the breathing affected by the sensor through it's a catheter structure?

Can the authors state the differences between the upper airway of human body and the pig, as the experiments were conducted on the animal. The performance of the sensor on the human body is not clear.

The authors should explain if the sensor provides the wrong signal when the human body do some action such as swallowing.

The discussion seems not enough. The advantages and disadvantages of this work are suggested to be highlighted. The results can be analyzed in detail. Can the authors compare this work with other current solutions in size, accuracy, safety, etc.

The experiments are not enough. Only the animal tests are conducted. The basic performance of this sensor is absent such as the stiffness.

Version 1:

Reviewer comments:

Reviewer #1

(Remarks to the Author)

The authors have thoughtfully addressed the reviewer comments, conducting a more extensive literature review of other techniques as well as the conduct of additional experiments. Specifically, the revised manuscript now contains a more detailed explanation of alternative approaches for assessing site of collapse, and the pros and cons of each relative to the new technology. The additional experiments have convincingly addressed queries about assessment of multiple sites of collapse and properties of the catheter related to comfort.

Reviewer #2

(Remarks to the Author)

The authors have addressed all my concerns.

Response to Reviewers' Comments

Manuscript ID NCOMMS-24-48321A titled "A Flexible Catheter-Based Sensor Array for Upper Airway Soft Tissues Pressure Monitoring"

Thank you very much for the prompt consideration and review. The major concerns from all reviewers are about the comfort of the device we proposed and the comparison with other similar approaches. We have supplemented some experiments and conducted a comprehensive literature review.

Please find detailed responses below to each point mentioned in the reviews. We have revised the manuscript accordingly and hope the revised version addressed all the concerns of both reviewers.

Reviewer #1:

Comment 1: Numerous previous attempts to assess the site of collapse using diverse techniques have been reported, and the authors should acknowledge such studies. These include studies of a multisensor catheter (doi.org/10.1093/sleep/29.5.666), optical coherence tomography (doi.org/10.1364/OE.11.001817), flow shape analysis (doi.org/10.1016/j.chest.2017.06.017), and snoring analysis ([10.1109/EMBC44109.2020.9175591](https://doi.org/10.1109/EMBC44109.2020.9175591)).

Response:

Thank you for your suggestions. Localization of the site of obstruction is important for accurate diagnosis and subsequent treatment of obstructive sleep apnea. We conducted a comprehensive literature review regarding the techniques for obstruction site locating. Multi-sensor catheters, optical coherence tomography (OCT), flow pattern analysis, ultrasound imaging, and snoring sound frequency analysis are also currently available in clinical studies for locating OSA obstruction sites.

Several techniques for OSA obstruction site localization have been proposed in the literature, including multi-sensor catheters, optical coherence tomography (OCT), flow pattern analysis, local ultrasound, and snoring sound frequency analysis. While these methods conducted promising attempts from various technical routes, certain limitations still hindered their performances. Multi-sensor catheters that enter the esophagus through the upper airway primarily detect changes in air pressure in different parts of the respiratory process, rather than measuring soft tissue contact pressure directly. Multi-sensor catheters with FBG technology can measure airway contact pressure gradients accurately, but their protective coating and cable jacket may affect flexibility and patient comfort. OCT provides real-time 3D airway imaging but is limited by respiratory movement interference and high cost. Local ultrasound is effective for real-time soft tissue monitoring but struggles with deeper structure measurement and high operator skill dependency. Non-invasive flow patterns and

snoring sound analysis are indirect measurements with low accuracy and spatial resolution, particularly in multiple obstruction site identification. Snoring sound analysis is also easily affected by external factors like background noise and patient positioning.

We have added a description and analysis of these methods to the discussion section of the manuscript. These modifications can be found on pages 11-12, lines 406-420.

Comment 2: One aspect that is relevant to human OSA is that there are usually multiple sites of airway collapse. The authors should address the question of whether their approach is able to detect multiple simultaneous sites of collapse. Or does complete collapse at one site prevent detection of collapse at other sites?

Response:

We appreciate the reviewer's insightful comment. Our sensor system is capable of detecting multiple simultaneous sites of airway collapse. As demonstrated in our experimental results, the obstruction locating capability of the pressure sensor array was investigated by comparing the position of the stenosis area in the CT image with the pressure response measured by the sensor array. **Supplementary Fig. 8** is the result of the experimental group of pigs. The position of the minimum cross-sectional area in the upper airway CT reconstruction model was compared to the pressure distribution heatmap generated by the sensor array, with the epiglottis serving as the reference plane. During this experiment, Pig 4 exhibited multiple obstruction sites along the airway on the modeling day, which was effectively identified by our sensor arrays. We provide a schematic of Pig 4 in **Supplementary Fig. 9a**, illustrating the correlation between the sensor-detected collapse locations and the corresponding CT results. This diagram highlights the ability of our system to accurately detect multiple simultaneous airway obstructions.

The injection of a sodium hyaluronate mixture into the soft palate and lateral pharyngeal wall of Bama pigs during the establishment of an OSA animal model resulted in stenosis of the upper airway. The application scenarios of the sensor array on the No. 4 OSA model pig and the corresponding CT sagittal images are shown in **Supplementary Fig. 9 a, b**. The location of the stenosis area in the upper airway CT reconstruction model was compared to the pressure distribution heatmap generated by the sensor array, with the epiglottis serving as the reference plane (**Supplementary Fig. 9 c**). On the day of modeling, CT imaging revealed two distinct stenotic areas within the airway, exhibiting a "double peak" characteristic with a total length of 26.9 mm (**Supplementary Fig. 9 b, c**). The heat map generated by the sensor array on this day demonstrated high contact pressure values for Ch4 and Ch8, with local pressures exceeding 15mmHg, while other sensor pressures remained below 10mmHg. Based on sensor unit distribution, Ch4

covered a sensing range from 30 to 36mm from the front of the catheter, whereas Ch8 had a sensing range from 54 to 60mm. Consequently, Ch4 to Ch8 collectively covered a total sensing area length of approximately 30mm. These measurement results confirmed that both location and length characteristics observed through CT imaging corresponded with pressure distribution depicted by the heat map generated by the sensor array, thus validating the ability of our system to detect multiple simultaneous airway obstructions accurately.

We added the results (Supplementary Fig. 9) to the Supplementary Material and described them in the Results section of the manuscript. (see page 10, lines 341-342).

[figure redacted]

Supplementary Fig. 9 The application scenarios of the sensor array (a), upper airway CT image (b), and sensor pressure distribution heat map (c) in No.4 OSA model pig on the day of modeling.

Reviewer #2:

Comment 1: The authors provide a long-term solution for upper airway soft tissues pressure monitoring. How about the comfort and safety of this sensor? How about the flexibility of it?

Response:

To investigate the comfort and flexibility of the sensor array, we collected several clinical catheters currently used to enter the upper airway or esophagus through the nasal cavity. The catheters include silicone gastric tubes, polyurethane (PU) gastric tubes, polyurethane PH-impedance catheters, and PVC nasal trachea cannula. The tensile and bending modulus of all catheters were tested using a universal materials testing machine (EZ-LX HS, Shimadzu, Japan). The minimum size for clinical use was chosen as the dimensions of all devices to be tested.

The bending modulus plays the most important role in ensuring the comfort and safety of those devices when used in the airway. The results showed that the bending modulus of all the gastric tubes is <15 MPa (**Supplementary Table 1**). The bending modulus of the sensor array is 5.86 MPa, close to the lowest value (4.52 MPa) among all the detected devices. Meanwhile, the bending modulus of the PU gastric tube used as the sensor array substrate is 14.21 MPa. The decrease in the bending modulus of the sensor array compared with that of the PU gastric tube may be related to the femtosecond laser etching process on the surface of the tube. In terms of tensile modulus, the tensile modulus of all gastric tubes is <30 MPa, and the tensile modulus of the PH-impedance catheter (34.67 MPa) and nasal trachea cannula (87.92 MPa, 90.06 MPa) is significantly

higher than that of the gastric tubes and sensor array. The tensile modulus of the sensor array (26.13 MPa) remained at a similar level to that of the PU gastric tube (22.17 MPa). The bending and tensile modulus results show that the sensor array offers similar comfort during use as nasal catheters adapted for clinical settings.

In terms of safety, the low bending modulus reduces the risk of damage to the upper airway tissue by the device. To verify the safety of the devices, we set up a control group in which no sensor array was placed in the upper airway of the OSA model pig. Inflammation-related indexes in serum and upper airway secretions of pigs from both groups were examined at various time points in the experiment, and the experimental and control groups had the same trend (Figure 4c-h). The biocompatibility of the catheter-based flexible pressure sensor array was also validated through 24-hour placement in a pig’s upper airway (Figure 4i-l). Histopathological examination showed normal tissue structure with no abnormal fibrous connective tissue or inflammation, consistent with the untreated control pig. No significant differences in collagen fiber content were observed between the two groups. These results confirm that the sensor array did not induce systemic or local inflammatory responses, demonstrating excellent biocompatibility.

The results are updated in the revised manuscript on pages 7-8, lines 258-265. Supplementary Table 1 has been added as a supplement.

Supplementary Table 1. Elastic Modulus of clinically used trans-nasal tubing devices

Item	Gastric tube 1	Gastric tube 2	Gastric tube 3	PH-Impedance Catheter	Nasal trachea cannula 1	Nasal trachea cannula 2	Sensor Array
Material	Silicone	Silicone	Polyurethane	Polyurethane	PVC	PVC	Polyurethane Base + PDMS + Polyurethane Film

Outer Diameter (mm)	2	2	2	2	3	3	2.6
Inner Diameter (mm)	1.2	1.2	1.2	1.4	2	2	1.2
Bending Modulus (MPa)	10.90	4.52	14.21	47.84	39.21	44.83	5.86
Tensile Modulus (MPa)	22.4	6.21	22.17	34.67	87.92	90.06	26.13

Devices manufacturer:

Gastric tube 1: Yangzhou Huayue Technology Development Co., Ltd. HUAYUE Gastric Tube. Yangzhou, China.

Gastric tube 2: Huizhong International Medical Devices Co., Ltd. VREPER Gastric Tube. Beijing, China.

Gastric tube 3: Nutricia Pharmaceutical (Wuxi) Co., Ltd. Flocare Nasogastric Tube. Wuxi, China.

Sensor Array: This work.

PH-Impedance Catheter: Chongqing Jinshan Science & Technology (Group) Co., Ltd. Jinshan pH-Impedance Catheter. Chongqing, China.

Nasal trachea cannula 1: Henan Tuoren Medical Instrument Group Co., Ltd. TUOREN Tracheal Tube. Henan, China.

Nasal trachea cannula 2: Jiangxi Glance Medical Equipment Co., Ltd. GLANCE MEDICAL Tracheal Tube. Jiangxi, China.

Comment 2: Are there any other similar solutions for soft tissue pressure monitoring?

Response:

Catheter-based flexible pressure sensors have numerous application scenarios in vivo, particularly for vascular and human body cavity soft tissue pressure measurement. I summarize the main application scenarios for similar soft tissue pressure monitoring solutions in clinical trials.

1. Vascular Pressure Monitoring

Vascular pressure measurements integrated into catheters and balloons have appeared in several studies. John A. Rogers' team has developed multiparameter sensing arrays that can be integrated on the outer surface of endocardial balloon catheters for high-density spatial mapping of temperature, pressure, and electrophysiological parameters and allow for programmable electrical stimulation. Kang et al. developed a balloon catheter-integrated piezoelectric micro-pyramid array sensor (p-MPB) to measure vascular stiffness for the early diagnosis of atherosclerotic symptoms.

2. Urological Soft Tissue Pressure Monitoring

Catheter-based pressure sensors are commonly used in urodynamics to measure pressure in the bladder and urethra to assess conditions such as urinary incontinence (UI) and bladder dysfunction. M. Ahmadi groups utilize super-capacitive sensors and capacitive sensor strips, providing high sensitivity and real-time measurements, enabling clinicians to detect pressure variations along the urethra.

3. Gastrointestinal soft tissue pressure monitoring

Catheter-based pressure sensors are widely used to monitor esophageal and gastric motility disorders, facilitating the diagnosis of conditions such as dysmotility and gastroesophageal reflux disease (GERD). Traditional methods include high-resolution manometry (HRM), which employs multiple pressure sensors distributed along the catheter to measure esophageal muscle contractions. An example is the VIPUN Gastric Monitoring System, which incorporates a balloon catheter integrated with a pressure sensor to measure intragastric pressure, gastric contractility, and motility in real time, allowing clinicians to effectively identify gastric motility disorders.

The above approaches for soft tissue pressure monitoring have been summarized and added to the introduction section of the manuscript on page 4, lines 121-123.

Comment 3: Is the breathing affected by the sensor through its a catheter structure?

Response:

We thank the reviewer for the insightful question. Ensuring the comfort of the device while minimizing its impact on breathing is critical. To avoid the impact of the device on the patient's breathing, the substrate used for the sensor array is a medical polyurethane gastric tube with a diameter of 2 mm, specifically designed for infants ((in contrast, adult gastric tubes typically have a diameter of 5-8 mm). The outer diameter of the sensor array is 2.6 mm. The tensile modulus of the sensor array is close to that of polyurethane gastric tubes, and the bending modulus is lower than that of polyurethane gastric tubes (detailed information can be found in **Supplementary Table 1**), which provides better comfort. These characteristics contribute to enhanced comfort and minimize the effect on breathing during use.

Comment 4: Can the authors state the differences between the upper airway of the human body and the pig, as the experiments were conducted on the animal? The performance of the sensor on the human body is not clear.

Response:

Thank you for your insightful question regarding the differences between the upper airway of humans and pigs, and how this may affect the performance of the sensor on the human body. Below, I have outlined the key similarities and differences between

the two, as well as addressed the potential implications for the sensor's performance in humans.

Similarities:

Anatomy: Humans and pigs have similar upper airway structures, including the nasal cavity, soft palate, pharynx, and larynx. These similarities make the pig a suitable model for studying upper airway pathology and testing devices such as pressure sensors.

Soft tissue function and histology: Both pigs and humans have airway walls composed of collapsible soft tissues in the upper airway. Collapse of these soft tissues can obstruct airflow during respiration, which is critical for studying the development of diseases such as OSA. Both airway walls have similar histological structures, and histopathological analyses are consistent under the same conditions.

Respiratory physiology: Despite differences in respiration rates, pigs and humans breathe by moving the diaphragm, making pigs a good model for studying airway obstruction and pressure changes.

Discrepancy:

Upper airway length and geometry: Although pig airway size is closer to that of humans, the pig upper airway is longer, and the curvature of the soft palate posterior region is less. Compared to humans, pigs have a higher epiglottis position, which contacts the soft palate and separates the oral cavity from the nasopharynx. These geometrical differences may affect the position of the sensor in contact with upper airway structures.

Respiratory rate and pattern: Pigs have a higher respiratory rate than humans, which may influence the sensor's response to pressure variations. In humans, the sensor is expected to detect slower breathing cycles and larger tidal volumes.

These differences, along with the limitations of animal studies, will be addressed in the Discussion section of our manuscript on page 12, lines 436 -440. Nevertheless, using the Bama pig model, particularly with targeted soft palate region obstruction, allows us to simulate human OSA conditions, laying a solid foundation for future human trials.

Comment 5: The authors should explain if the sensor provides the wrong signal when the human body do some action such as swallowing.

Response:

To detect respiratory-related behaviors beyond obstructive sleep apnea, we compared

video data with polysomnography (PSG) results. We extracted snoring and mastication-deglutition data from the animal experimental results (**Supplementary Fig. 6** and **Supplementary Fig. 7**). As illustrated in Supplementary Fig. 6, during snoring, the soft tissue of the upper airway in pigs cyclically applies and releases pressure on certain units of the catheter-based sensor, corresponding to the snoring frequency. This process is periodic, similar to normal breathing, but the peak pressure exerted by the upper airway soft tissue during snoring is greater than that during normal breathing, which is reflected in the increased pressure amplitude of the sensor. During snoring, which can act as a precursor to obstructive sleep apnea, the airway tissue pressure gradually rises with the persistence of snoring, leading to a slow decline in the baseline of certain sensing units, such as Ch7.

Supplementary Fig. 7 illustrates the PSG and sensor array signals during chewing and the subsequent swallowing event, which is indicated in the highlighted section. During chewing, the pig exhibits slow, unobstructed, and notably shallower breathing, as evidenced by longer and lower amplitude signals compared to the preceding regular breathing cycles. Following the period of chewing, the experimental pig initiated a swallow action, which resulted in a transient compression and release of the sensor catheter, resulting in a significant waveform peak in the sensor signal.

By analyzing the signal characteristics of the sensing array during snoring, chewing, and swallowing in experimental pigs, we identified distinct pressure measurement features corresponding to each event. This finding demonstrates that while monitoring upper airway pressure during sleep, other human movements do not generate waveforms that could be misinterpreted as the target signal associated with obstructive sleep apnea. This specificity enhances the reliability of the sensing array for accurately distinguishing obstructive sleep apnea from other respiratory events.

To respond to comments, we added the results (Supplementary Fig. 6 and 7) to the Supplementary Material and described them in the Results section of the manuscript. (see page 9, lines 304-309).

Supplementary Fig. 6 PSG and sensor array data during snoring in OSA model pigs.

Supplementary Fig. 7 PSG and sensor array data during chewing and swallowing events in OSA model pigs.

Comment 6: The discussion seems not enough. The advantages and disadvantages of this work are suggested to be highlighted. The results can be analyzed in detail. Can the authors compare this work with other current solutions in size, accuracy, safety, etc.

Response:

Thank you for your insightful comment. In this response, we have expanded the discussion to highlight both the advantages and disadvantages of our work and provided a detailed comparison with other current solutions based on factors such as size, accuracy, safety, and invasiveness (Supplementary Table 2).

Advantages of this work:

1. In-situ pressure measurement: This study introduces a catheter-based flexible pressure sensor array that continuously monitors soft tissue pressure in the upper airway, enabling high-resolution obstruction localization. This sensor array provides direct in situ contact pressure distribution data for airway obstruction.
2. Processing methods and device characteristics: This novel approach incorporated an advanced 3D femtosecond laser fabrication process that forms functional microstructure directly on the catheter to realize pressure sensing function for the upper airway with high spatial resolution, excellent stability, high sensitivity (38.1 Ω /mmHg), and rapid response time (660 ms), making it highly effective for real-time obstruction localization. The bending modulus of the sensor array is 5.86 MPa, and the tensile modulus is 26.13 MPa. These values indicate that the sensors maintain sufficient flexibility to minimize discomfort during use.
3. In vitro and in vivo experimental validation: We validated the performance of our sensor array through in vitro tests and in vivo experiments using a reversible OSA pig model. The results showed precise detection and localization of obstructions, consistent with polysomnography data and CT imaging, underscoring the potential of our technology for accurate OSA obstruction site locating. Our sensing arrays can locate obstruction at the millimeter level at multiple locations.

Disadvantages:

1. Reducing the device dimension to further increase comfort will be the direction of our future research.
2. Although we validated the performance of the sensors on a reversible OSA animal model that verified the sensor's performance under controlled conditions. There are still differences between real human OSA patients and animal models, which will pose a challenge in the clinical use of the sensor array.

We added the method comparison results (Supplementary Table 2) to the Supplementary Material and described them in the Discussion section (see pages 11-12, lines 405 -434).

Supplementary Table 2 Different solutions for locating OSA obstruction sites

Method	Sedation Required	Invasive/non-invasive	Direct/Indirect Measurement	Device Size	Locating accuracy	Additional safety requirement
Polysomnography (PSG)	Yes (for some)	Invasive	Indirect	Diameter 2mm (sensor module)	A sensor module is required to determine if the airway is obstructed by comparing air pressure differences.	
Flexible Laryngoscopy	No	Invasive	Direct	Diameter 3-5mm (laryngoscopy)	Visual examination	Need local anesthetic
Multi-Sensor Catheters with FBG	Yes (for some)	Invasive	Direct	Diameter about 2-3mm	It can accurately measure the pressure gradient along the airway during sleep but is poorly flexible.	Sometimes need a local anesthetic
Optical Coherence Tomography (OCT)	No	Invasive	Direct	Diameter 1.2-3.8 mm(catheter)	High resolution for anatomy, but accuracy can be compromised by interference from respiratory movements and soft tissue collapse.	
Drug-Induced Sleep Endoscopy (DISE)	Yes	Invasive	Direct	Diameter 3-5mm (endoscope)	Dynamic imaging during sleep	Need local anesthetic
Pharyngoesophageal Manometry	No	Invasive	Indirect	Diameter 2-5 mm(catheter)	Low resolution	

Acoustic Pharyngometry	No	Non-invasive	Indirect	Probe diameter 10-15 mm	It cannot directly measure dynamic airway collapse and only provides anatomical information during static breathing.	
Rhinometry	No	Non-invasive	Indirect	Probe diameter 10-15 mm	It cannot directly measure dynamic airway collapse and only provides anatomical information during static breathing.	
Local Ultrasound	No	Non-invasive	Direct	Probe diameter 20-50 mm	Unable to accurately locate airway obstruction in multiple locations	
Snoring Sound Frequency Analysis	No	Non-invasive	Indirect	N/A	The accuracy can be affected by multiple interfering factors. Unable to accurately locate airway obstruction in multiple locations	
Flow Pattern Analysis	No	Non-invasive	Indirect	N/A	Unable to accurately locate airway obstruction in multiple locations	
Computed Tomography (CT)	No	Non-invasive	Direct	Scanner	High for anatomy, but no functional information	Ionizing radiation risk
Cine-MRI	Yes	Non-invasive	Direct	Scanner	Real-time imaging	
Catheter-Based Sensor Arrays (This work)	No	Invasive	Direct	Diameter 2.6 mm	It allows direct on-site pressure measurements, localizes obstructions to the millimeter level, and detects airway obstructions in multiple locations simultaneously.	

Comment 7: The experiments are not enough. Only the animal tests are conducted. The basic performance of this sensor is absent such as the stiffness.

Response:

Thank you for your insightful feedback. We have conducted additional experiments to provide more information on device flexibility.

We have conducted measurements of the bending and tensile moduli of the sensor array to enhance the understanding of the device's mechanical properties. As detailed in the newly added **Supplementary Table 1**, the sensor array has a bending modulus of 5.86 MPa and a tensile modulus of 26.13 MPa. These values fall within the range of commonly used trans-nasal devices, such as gastric tubes and nasotracheal cannulas, suggesting that the sensor possesses adequate flexibility to minimize discomfort during use while ensuring sufficient structural integrity.